# Electroencephalography (EEG)-Based Comfort Evaluation of Free-Form and Regular-Form Landscapes in Virtual Reality

**Hongguo Ren [1], Ziming Zheng [1], Jing Zhang [1,\*], Qingqin Wang [2] and Yujun Wang [1]**

[1] International Research Center of Architecture and Emotion, Hebei University of Engineering, Handan 056009, China; renhongguo@hebeu.edu.cn (H.R.); bonjourlapluie@outlook.com (Z.Z.); 15312889667@163.com (Y.W.)

[2] The State Key Laboratory of Building Safety and Built Environment, China Academy of Building Research, Beijing 100013, China; wangqq@cabr.com.cn

\* Correspondence: zhangjing01@hebeu.edu.cn; Tel.: +86-186-5920-1726

**Abstract:** Urban landscape parks play a crucial role in providing recreational opportunities for citizens. Different types of landscapes offer varying levels of comfort experiences. However, the assessment of landscape comfort primarily relies on subjective evaluations and basic physiological measurements, which lack sufficient quantification of relevant data. This study employed electroencephalography (EEG) technology and subjective questionnaire evaluation methods. Participants observed two sets of landscape demonstration videos using virtual reality (VR) devices, and EEG alpha values and subjective evaluation scores were collected to assess the comfort levels of free-form landscape and regular-form landscape. Additionally, this study explored the correlation between landscape characteristics and physiological comfort. The analysis of the results showed that: 1. The average amplitude of EEG alpha waves recorded from 11 electrodes in the left temporal lobe and right parietal lobe of the participants was higher after they watched the free-form landscape demonstration. The increased alpha values suggest that free-form landscapes are more likely to induce physiological comfort in these specific brain regions. In contrast, regular-form landscape was found to induce higher alpha values at seven specific electrodes located in the occipital cortex, right temporal lobe, and central regions of the participants. In general, free-form landscape provided physiological comfort to a greater number of brain regions. 2. The two groups of landscapes exhibit distinct subjective cognitive differences in terms of their landscape characteristics. These differences, ranked in order of magnitude, include rhythmicity, sense of order, sense of security, and sense of dependence. 3. This study examined the $\alpha$-waves of specific brain regions, including the right and left temporal lobe and occipital lobe, as well as subjective scoring. It discovered that the rhythmicity, degree of variation, degree of color, and sense of nature of a landscape impact the $\alpha$-wave value of electrodes in different brain regions. Moreover, there exists a certain linear relationship between the four landscape features and the $\alpha$-wave values in different regions of the brain. The results of this study provide some reference for the creation of a comfortable landscape design.

**Keywords:** free-form landscape; regular-form landscape; virtual reality (VR); EEG; alpha value; landscape characteristics; comfort level

## 1. Introduction

### 1.1. Research Background

The city park plays a crucial role in providing people with opportunities for leisure and recreation, while also exerting a significant influence on their physical and mental well-being. Following the COVID-19 epidemic, there has been a gradual rise in people's desire for garden green spaces, leading to an increasing variety of themes in park development. Based on the description by British gardener G.A. Jellicoe, the world garden can be categorized into three major genres: China, West Asia, and Ancient Greece. From the formal point of view, Chinese garden roads are more free and hidden, and plant growth is more natural;

this form of garden is very common in southern China [1–3]. Western garden plants are neatly pruned and their appearance is of geometric form with a pursuit of straight lines and the feeling of order, especially in the 18th century in the British landscape parks [4–6]. The Ancient Greek garden style is, to some extent, a combination of the former two: in the core part of the garden is usually presented a regular geometric form but, in the garden boundary, processing is more natural. Overall, the world's gardens are regular, natural, and mixed, and although this categorization may ignore the subtle differences in regional styles, it still provides a suitable basis for the study of garden layout patterns [7]. Although urban gardens today have a wide variety of themes, their layout is still predominantly based on these three main types. However, there is currently limited research on the impact of different landscape layouts on the comfort experienced by park visitors. Most previous evaluations of park landscape comfort have been subjective and based on cognitive assessments, with relatively little quantitative analysis using physiological data indicators to measure human comfort. In this experiment, we utilized electroencephalography (EEG) to gather brain wave data from participants as they viewed videos of regular-form landscapes and free-form landscapes. The brain wave data were collected using a brain electrode device, and the alpha value of the brain wave was extracted as a measure of physiological comfort. This method allowed us to quantify the level of physiological comfort experienced by the participants. Additionally, we used a subjective questionnaire to evaluate the subjective characteristics of the two types of landscapes, which aided in our study.

The objective of this study was to assess the disparity in the impact of regular- and free-form landscapes on human comfort, specifically focusing on the electroencephalogram (EEG) perspective. Additionally, this study aimed to examine the variations in the effects of different landscape characteristics on the comfort levels of different brain regions. The EEG technology accurately captured the alterations in the respondents' physiological data, while the subjective questionnaire data elucidated the circumstances that triggered the corresponding physiological changes.

### 1.2. EEG Signals and Related Theoretical Research

The human body uses its physiological organs to receive external information and transmit it to the brain. Even small stimuli can be detected in the brain waves. The frequency of human brain waves generally ranges from 1 to 30 Hz. These waves can be divided into five bands based on their frequency: delta wave $\delta$ (0.5–3 Hz), theta–delta wave $\delta$ (0.5–3 Hz), theta wave $\theta$ (4–8 Hz), alpha wave $\alpha$ (8–13 Hz), beta wave $\beta$ (14–30 Hz), gamma wave $\gamma$ (>30 Hz), and mu wave $\mu$ (9–11 Hz) [8]. Among these brain waves, the $\delta$-wave signifies sleep, fatigue, and the subconscious state. The theta wave indicates sleepiness, deep relaxation, and the subconscious state. The gamma wave represents higher cognitive activity, which is a result of heightened neuron excitability. The alpha wave, with a frequency range of 8–13 Hz, acts as a "frequency bridge" connecting the conscious mind (beta) and the subconscious mind (theta). The $\alpha$-wave is associated with inducing a state of calmness and deep relaxation in the human body. It is commonly observed during periods of relaxation, tranquility, and wakefulness without stress. On the other hand, the $\beta$-wave is typically observed during alertness, cognitive activities, and focused thinking [9,10]. Therefore, we utilize the $\alpha$-wave to determine if the human body is in a state of comfort and calmness: the higher the $\alpha$ value, the more preferable it is for the human body to be in a comfortable state [8,11]. This study introduced EEG alpha waves to quantify human comfort while incorporating subjective questionnaire evaluations hoping to further investigate the relationship between park landscape features and comfort.

Multiple researchers have undertaken experiments that examine the correlation between EEG and the perception of landscapes. Linhong Wang conducted a study to investigate how the color of highway landscapes affects drivers' EEG $\alpha$ value. The results revealed that landscape color has varying degrees of impact on drivers' EEG $\alpha$ value. Furthermore, the average value of landscape color is negatively correlated with drivers' EEG $\alpha$ value. As the landscape color and vividness increase, the $\alpha$ value of drivers tends to approach 1,

indicating a state of equilibrium between relaxation and vigilance [12]. Jun Qin conducted an experiment where physiological indicators, such as human EEG cardiovascular, were measured by manipulating the characteristics of plants, such as color, smell, and size. The results of the EEG analysis revealed that the relative power of the EEG $\alpha + \beta$ bands increased when participants were exposed to green plants, indicating a state of comfort [13]. Ahmad Hassan conducted a study on the psychological and physiological effects of adolescents walking in bamboo forests versus urban environments. The findings revealed that adolescents walking in bamboo forest environments exhibited significantly higher levels of alpha waves compared to those in urban environments. Additionally, the participants reported feeling more relaxed, comfortable, and at ease, and experienced reduced anxiety after walking in bamboo forests, as indicated by the questionnaire results [14]. Roger S. Ulrich conducted an experiment where he measured the brain waves of participants while they were looking at different types of landscapes: natural and urban. The results revealed that when participants were looking at natural landscapes, their brain wave alpha values were significantly higher compared to when they were looking at urban landscapes. Additionally, the mean alpha values were higher when participants were viewing water features. These findings suggest that natural landscapes and water features have a positive impact on human mood [15]. In addition to visual landscapes, scholars have also conducted research on other sensory aspects, such as acoustics using electroencephalography (EEG). For instance, Heng Li investigated the perception of acoustic components in typical mountainous areas and urban parks using EEG recordings [16].

In addition to the research on landscapes and EEG, scholars have also conducted relevant studies on the relationship between spatial perception and EEG. Lemya Kacha conducted EEG physiological measurements on street scene perception. The results indicated significant changes in the alpha and beta power bands in the parietal and frontal lobes during street scene perception. Furthermore, as familiarity with street scene images increased, there was as higher increase in alpha power, reflecting a higher level of relaxation [17].

Using EEG, Sun Xia discovered that the average alpha wave value is closely related to the perception of spatial openness and closure in commercial districts. Additionally, the perception of regional spatial scale and color have a direct influence on the generation of alpha waves, as indicated by the SD factor scores, picture composition proportion, and the correlation between color and alpha waves, as observed during the research. Commercial avenues characterized by minimalist color schemes and enclosed environments have the potential to enhance alpha brainwave activity, hence inducing a heightened sense of relaxation and tranquility in individuals [18].

In general, landscape research samples that utilize EEG technology primarily rely on pictures, which restricts the subjects' field of view and results in a relatively low level of realism in the simulation. Additionally, most of the existing research focuses on physical characteristics such as size, color, and shape, and their impact on human comfort. However, there is limited exploration of the relationship between landscape layout characteristics, such as brightness, enclosure, and rhythmicity, and their influence on comfort. Thus, this study utilizes EEG technology and employs virtual reality (VR) simulation to generate an immersive landscape encounter through a 360° video presentation. Finally, mathematical analysis and subjective questionnaires were used to evaluate the comfort level of the two groups of scenarios as well as to explore the relationship between landscape features and comfort level.

## 2. Experimental Methods and Design

This study employed EEG analysis, subjective questionnaire, and SPSS mathematical analysis. 1. The EEG signals of the participants were captured using EEG equipment while they viewed demonstration videos from two different program groups. The corresponding EEG alpha waves were then extracted and compared to investigate the impact of the two program groups on the subjects' comfort level. 2. The researchers utilized a subjective questionnaire to carefully choose 15 pertinent questions and gather the participants' subjective

responses. SPSS was used to establish the correlation between subjective ratings and EEG alpha values and study the connection between landscape features in subjective evaluation and EEG alpha values.

### 2.1. Design and Experimental Video

The purpose of this experiment was to study the difference in physiological comfort of the brain when subjects watch regular and freestyle landscapes, and to understand changes in physiological comfort by recording the EEG data of subjects when they watch video demonstrations. In landscape gardening and related professions, the distinction between free-form landscape and regular landscape has been established at a theoretical level. Generally, the regular landscape exhibits a strong axial relationship, with straight roads and consecutive geometric spaces. Other landscape elements also exhibit a certain geometric form. On the other hand, the free-form landscape lacks a clear axial relationship, with curved roads and a more random distribution of space and greenery. The landscape vignettes are dispersedly arranged. Based on the applicable theories and definitions, we chose a combined area of 14,500 square meters for the design of both the free-form (Design D1) and regular (Design D2) landscape virtual models (see Tables 1 and 2). In our design, the free-form landscape lacks a clear axial relationship and has an indistinct spatial layout, characterized by a zigzagging path and a more natural arrangement of greenery. The vegetation in this area is tall and exhibits darker hues (see Table 3). The regular landscape exhibits a clear east–west alignment, with spatial arrangements displaying notable consistency and rhythm. The spaces are interconnected by straight paved roads, while the layout of plants in the landscape adheres to a regular pattern. The landscape vignettes showcase regular geometric shapes, and the overall color scheme of the landscape is vibrant (see Table 4).

**Table 1.** Design D1 general floor plan and browsing route.

**Route and Node Location**

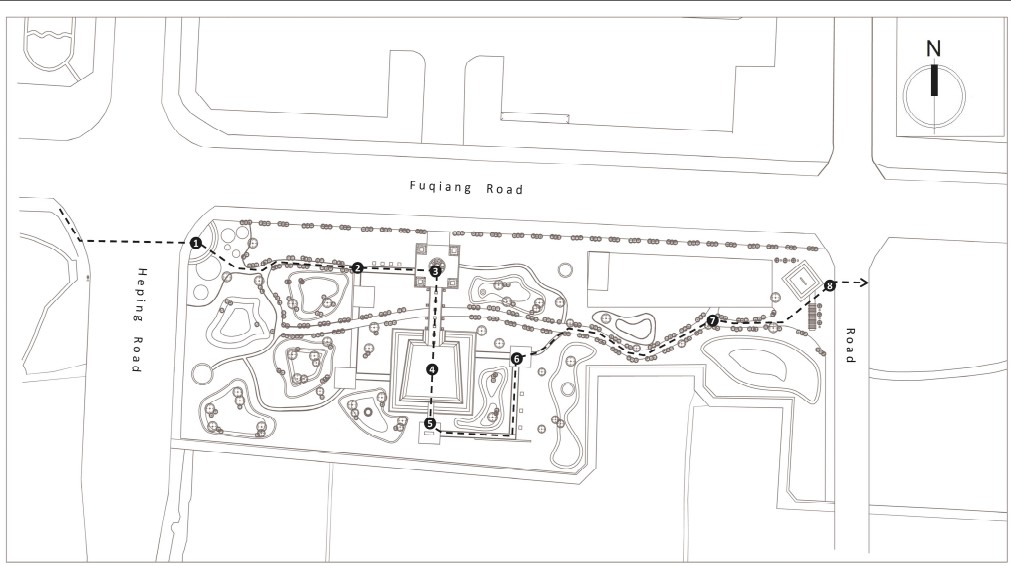

**Table 2.** Design D2 general floor plan and browsing route.

**Route and Node Location**

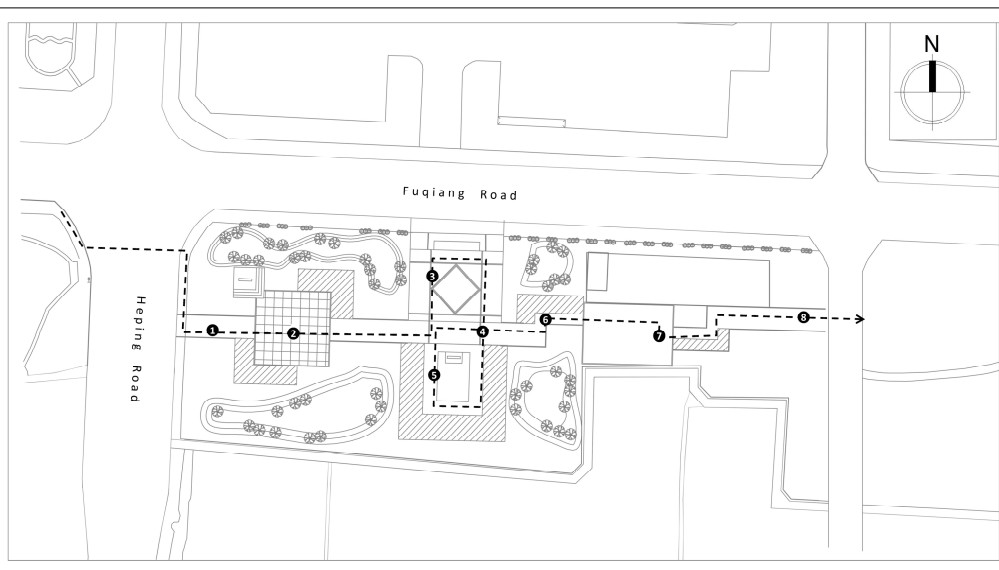

**Table 3.** Partial effect displays of plan D1.

| Location 1 | Location 2 | Location 3 | Location 4 |
|---|---|---|---|
| | | | |
| Description: | | | |
| West entrance: no axis relationship, sets up a modern sense of landscape sketches. | Small square: strong concealment, unilateral arrangement of landscape walls. | The main entrance on the north side: a relatively open central plaza with a certain axis relationship. | The large square on the north side: the scale is larger, the closure is stronger, and it is surrounded by a landscape wall. |
| Location 5 | Location 6 | Location 7 | Location 8 |
| | | | |
| Description: | | | |
| The small square on the south side: strong concealment and landscape sculptures. | Small square: surrounded by trees, strong privacy. | Park trails: There is greenery on both sides of the road. | East entrance: open plaza with landscape sculptures. |

**Table 4.** Partial effect displays of plan D2.

| Location 1 | Location 2 | Location 3 | Location 4 |
|---|---|---|---|
| | | | |
| **Description:** | | | |
| West entrance: obvious axis relationship, no landscape sketches. | Small square: The openness is strong, and sculptures and landscape walls are arranged in the corners. | The main entrance on the north side: a relatively open central plaza with a certain axis relationship. | The large square on the south side: the scale is suitable for a relatively closed area, surrounded by a landscape wall on three sides. |
| **Location 5** | **Location 6** | **Location 7** | **Location 8** |
| | | | |
| **Description:** | | | |
| The large square on the south side: the scale is appropriate, and the landscape sketches are arranged. | Small square: more spacious | Park trails: large width, single-sided landscape greenery, and arrangement of covered landscape. | East exit: wide roadway, no sculpture arrangement. |

We created virtual models of free-form landscape and regular-form landscape by using Sketch Up software (3D Design SoftwareSketchUp Pro 2023) and chose the MARS rendering software (Chongqing Guanghui City Technology Co., Ltd., Chongqing, China, 2020), which is a more realistic simulation of the environment, to render the virtual model and produce the experimental demonstration video, which was produced by the same person for the two groups of program videos (Figure 1). Before the demonstration video production, we selected 8 important landscape nodes in each of the two groups of programs in order to ensure that the demonstration videos of both landscape design plans show these important nodes. To ensure that the two video playback speeds are not significantly different, we stipulated that the paths of the two groups of video demonstrations should travel from the west to the east, where the path of D1 design is 542 m, and the D2 plan is 536 m.

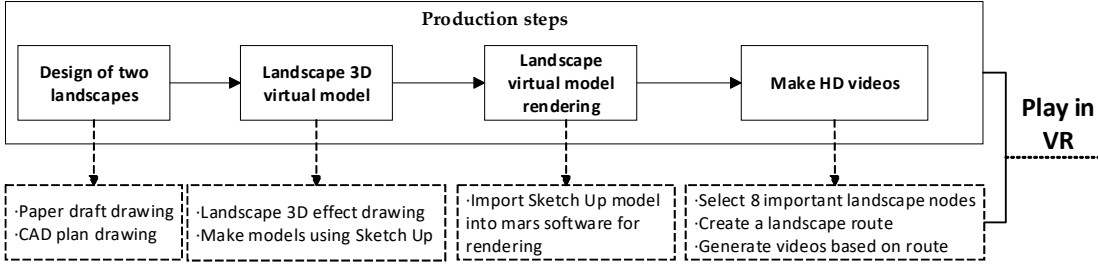

**Figure 1.** Landscape plan experimental video production process.

In the rendering software MARS, the ambient time was set to 16.00 on 21 June, with clear daytime as the real environment. The point of view of an adult male with a height of 178 was used as the view plane, and the demonstration video was generated at a normal walking speed according to the specified path (Scenario 1 1.39 m/s, Scenario 2 1.37 m/s). To maximize the authenticity of the visual stimulation of the video samples, the videos

used a 360° panorama at 60 frames, with a resolution of 2048*1080 (2K) pixels, the video durations were all 6 min and 30 s, and the final VR equipment was selected for playback.

Prior to the VR experiment, the subjects were instructed to engage in the experience to mentally prepare themselves and were advised to assume a comfortable posture to avoid discomfort or dizziness caused by the VR equipment and the 3D video demonstration. Subjects were permitted to gradually swivel their heads to survey their surroundings, thus preventing dizziness and minimizing disruptive motions that could impede the experiment. To mitigate the impact of individual physiological variables on the EEG data, the initial 30 s of the presentation video were allocated as a period of quiet adaptation. During this time, the baseline physiological data were recorded and then examined.

### 2.2. Experimental Materials

We selected VR all-in-one devices with millimeter-level positioning system and high-quality visual fidelity. The hardware parameters of the devices contain 4 K resolution, 110° field of view, 90 Hz refresh rate, Steam VR 2.0 tracking technology, and eye-tracking 110° viewable angle. In addition, the VR device supports the playback of panoramic HD videos produced by MARS software in Steam VR (HTC VIVE Pro, Wuhan Lingzhi Myriad Technology Company, Wuhan, China).

The electroencephalogram (EEG) equipment selected was a 64-channel electrode cap, with 32 electrodes chosen for recording at a sampling rate of 500 Hz. The EEG data were collected and recorded using SAGA software (see Figure 2). The electrode positions on the EEG cap were as follows: occipital lobe (Pz, Poz, O1, Oz, O2), left frontal lobe (Fp1, F7, F3), right frontal lobe (FP2, F4, F8), left temporal lobe (FC5, T7, C3), left parietal lobe (CP5, CP1, P7, P3), central (FC1, FC2, Cz, Fz), right temporal lobe (FC6, C4, T8), right parietal lobe (CP2, CP6, P4, P8), M1, M2, FPz.

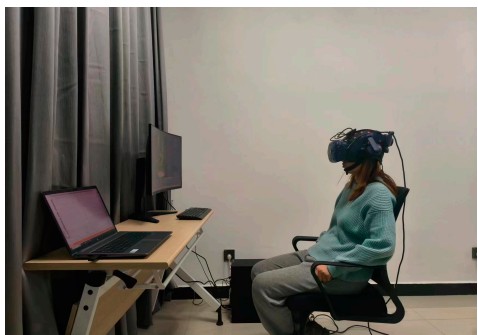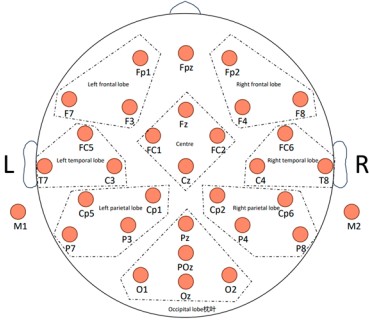

**Figure 2.** Experimental equipment and brain electrode distribution.

Experimental site selection: To minimize interference from factors other than the experimental design itself, a specific room within the Architecture and Sensory Engineering Laboratory of the School of Architecture and Art was chosen as the experimental site. The room has dimensions of 3 m in length and 2.4 m in width. It is equipped with blackout curtains to block external light sources. The room maintains a constant temperature of around 25 °C and is illuminated with LED diffused lighting. There is no other noise interference within the room (see Table 5).

Fifteen relevant issues were selected, including eight pairs related to physical sensations (space perception, brightness perception, scale perception, rhythm perception, order perception, variability, colorfulness, naturalness), four pairs related to psychological sensations (security, pleasure, interest, impression), and three pairs related to other questions (dwell ability, attractiveness, dependent). The questionnaire was administered in paper format, and participants were asked to rate their corresponding preferences on a scale ranging from −5 to 5.

**Table 5.** Experimental conditions.

| Items | Attribute |
|---|---|
| Location | Enclosed space |
| Temperature | 20° |
| Video | 6 min and 30 s 360° Panoramic 60 frame video |
| VR | VR all-in-one device |
| EEG devices | 64-channel lead-count electrode cap |
| Questionnaire | Paper questionnaire Paper questionnaire, 15 evaluation indicators, −5–5 points |

To make the 15 related questions easier to understand, we divided each question into five levels, each of which was described by a word, phrase, or sentence. For example, in the light perception question, we use "very dim", "relatively dim", "neutral", "relatively bright", and "very bright" to describe the five levels of brightness. We listed the 15 questions and their common descriptive words or sentences, labeled them with a range of scores, and finally printed the questionnaire in a horizontal format on A4 paper to facilitate scoring by the subjects themselves (see Table 6). We provided each subject with two questionnaires to prevent subjects from seeing the first scoring and then influencing the subsequent scoring.

**Table 6.** Content and format of the subjective questionnaire.

| Indicators Names | −5 Points | −2.5 Points | 0 Points | −2.5 Points | −5.0 Points | Any Score (−5–5) |
|---|---|---|---|---|---|---|
| Space perception | closed | relatively closed | neutral | relatively open | spacious | |
| Brightness perception | very dim | relatively dim | neutral | relatively bright | very bright | |
| Scale perception | loss of proportion | relatively imbalanced | neutral | relatively coordinated | proper proportion | |
| Rhythm Perception | no sense of rhythm | weak rhythm | neutral | strong rhythm | very strong sense of rhythm | |
| Order Perception | messy | relatively messy | neutral | relatively orderly | orderly | |
| Variability | very low sense of change | lack of variety | neutral | full of variety | very rich in changes | |
| Colorfulness | monotonous color | relatively monotonous | neutral | relatively rich | colorful | |
| Naturalness | very artificial | relatively artificial | neutral | relatively natural | very natural | |
| Security | very dangerous | relatively dangerous | neutral | relatively safe | very safe | |
| Pleasure | very melancholy | more melancholy | neutral | more pleasant | very pleasant | |
| Interest | very boring | relatively boring | neutral | relatively interesting | very interesting | |
| Impression | no impression at all | less impressive | neutral | more impressive | Very impressive | |
| Dwell ability | hope to leave immediately | want to leave | neutral | want to stay | hope to stay for a long time | |
| Attractiveness | very unattractive | less attractive | neutral | more attractive | very attractive | |
| Dependent | don't want to come again | probably won't come again | neutral | may come again | want to come very much | |
| **Name:** | | **Gender:** | | **Age:** | **Education** | |
| **Background:** | | | | | | |

### 2.3. Participant Selection

The experiment employed a within-subjects methodology to classify the physiological impacts of different landscape design styles on participants, dividing them into two con-

ditions: one after observing a free-form landscape and the other after observing a regular-form landscape. Participants were 30 students (12 male; age = 25 ± 5 years; 18 female; age = 25 ± 5 years) (see Table 7). The participants were selected from our student pool. Only those with a healthy central nervous system (CNS) and autonomic nervous system (ANS) state were included as participants. A sensitivity analysis of the sample size was performed using G*power. This analysis was performed using a within-subjects design *t*-test, with a significance level of 0.05 for both conditions (after watching the freestyle view and after viewing the regular view) and a statistical power of 0.8. Therefore, our sample can detect the effects of medium or large size. This study was approved by the Institutional Review Board of Hebei Engineering University (protocol code BER-YXY-2023031, approved on 10 June 2023), and the participants read and provided written informed consent.

**Table 7.** Basic information about experimental subjects.

| Item | Details | Unit: Person |
|---|---|---|
| Gender | Male | 12 |
| | Female | 18 |
| Age Group | 20–24 | 16 |
| | 24–28 | 13 |
| | 29–30 | 1 |
| Education Background | Bachelor's Degree | 3 |
| | Master's Degree and Above | 27 |

### 2.4. Experimental procedure

The experiment, which involved a substantial number of people, spanned six consecutive days from 15 September to 20 September 2023. Each day, the experiment was conducted during two time slots: from 2:00 p.m. to 6:00 p.m. and from 7:00 p.m. to 9:00 p.m. Participants were provided with information regarding the experimental protocol prior to the commencement of the experiment. Prior to commencing the experiment, the personnel equipped the participants with 64-bit electrode caps and simultaneously applied GT5 medical conductive paste to minimize impedance that may arise during the reading process; this ensures that the resistance of each electrode remains below 5 kilo-ohms. Following the EEG equipment setup, the subjects were required to wear VR all-in-one equipment. To achieve this, the See VR virtual reality eye-tracking device was used for eye calibration. The calibration process ensures that both eyes have a calibration rate of at least 80%. Once the subjects put on the experimental apparatus, they were allowed one minute to acclimate to the experiment. After ensuring that the subjects did not experience any noticeable discomfort, the video demonstration of design D1 commenced. During this demonstration, the experimenters assumed a comfortable seated position and were able to gradually rotate their heads for observation. The EEG waves were recorded using SAGA EEG analysis software (see Figure 3).

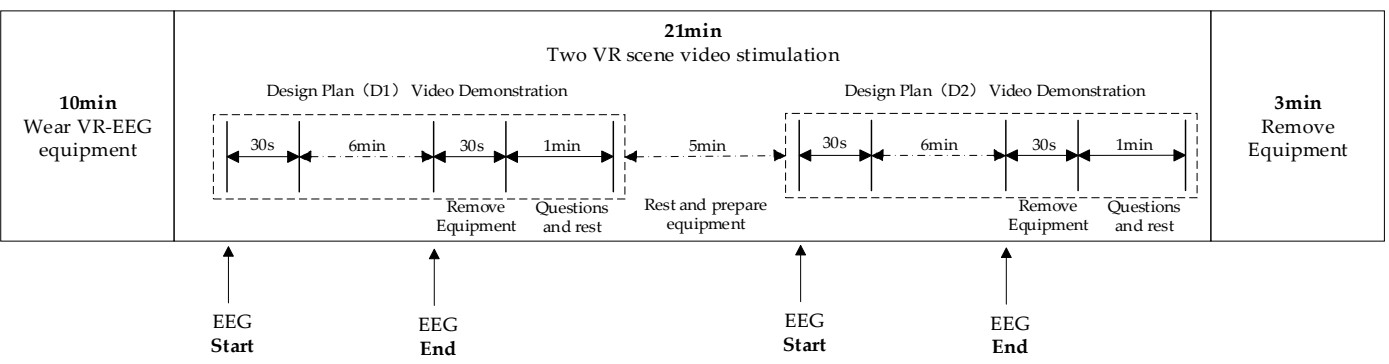

**Figure 3.** Experiment process.

Following the initial set of experiments, we promptly took away the participants' virtual reality (VR) devices within 30 s. Subsequently, the participants completed paper-based subjective questionnaires on the desktop within a one-minute timeframe, without any external interference in the experimental room during the scoring process. Following the completion of the scoring process, participants proceeded to a 5 min interval of relaxation, during which the personnel verified the proper functioning of the equipment. Once the patients finished resting and donned the VR equipment once more, the second set of situations was shown, concluding all the simulation studies. The entire experiment was carried out in a comfortable seated position to the greatest extent possible. To prevent interference from the external environment, all personnel not involved in the experiment were excluded. The experiment lasted around 35 min. To guarantee the precision and dependability of EEG signal extraction, novel participants utilized uncontaminated and dehydrated EEG caps, while other equipment was calibrated to the default values.

## 3. Results

### 3.1. Analytical Method

EEG Data Analysis: The alpha band (8–13 Hz) of the EEG signal is commonly observed in states of relaxation, tranquility, and non-stressful wakefulness, which is the required frequency range for this study. During the experiment, EEG signals were collected using SAGA software, and Python techniques were employed to extract and process the alpha wave amplitudes from each electrode in seconds. The relevant data were presented in the form of Excel spreadsheets. Subsequently, SPSS was used to analyze the differences in alpha values among the electrodes for conditions D1 and D2. Mean values and standard deviations were determined and other statistical analyses were performed to compare the impact of the two conditions on participant comfort.

Analysis of subjective evaluation results: The questionnaire results reflect the participants' subjective preferences and differences between conditions D1 and D2. Initially, the average scores and standard deviations of the factors in D1 and D2 were calculated using SPSS to gain preliminary insight into the participants' subjective inclinations. Subsequently, normality tests and analyses of variance were conducted to assess the statistical differences in factor scores between the two conditions. Finally, the corresponding results were analyzed to determine the subjective experiences and preferences of the participants.

Based on the analysis of EEG data and subjective ratings, a correlation analysis was performed between the significantly different alpha values of specific EEG electrodes and the subjective questionnaire scores, aiming to explore the relationship between landscape features and participant comfort.

### 3.2. EEG Analysis Results

In the EEG data analysis, the data are first pre-processed, which includes filtering, artifact removal, re-referencing, and feature extraction [Topological Features of Electroencephalography are Robust to Re-referencing and Preprocessing; EEG Integrated Platform Lossless (EEG-IP-L) pre-processing pipeline for objective signal quality assessment incorporating data annotation and blind source . . .]. Subsequently, statistical significance tests were employed to identify the eigenvalues of brain areas associated with free-form landscape (D1) and regular-form landscape (D2) (see Figure 4). After the experiment, we provided the subjects with a well-ventilated rest space to effectively reduce their fatigue.

#### 3.2.1. Individual Participant EEG Data Analysis

We compared the relationship between the magnitude of the mean alpha wave value per electrode for 30 individual subjects while watching the videos of the two schemes (see Table 8), and the results showed that, on 12 groups of electrodes, more than half of the subjects watched the free-form landscape video with a higher alpha value than they watched the regulated landscape video, 7 groups of electrodes did not have a significant difference in the alpha value, and 13 groups of electrodes tended to be in favor of the

regular-form landscape. From the above, both free-form and regularized landscapes can produce more α-waves on a larger number of electrodes and are more capable of eliciting physiological comfort. However, we further found that the two landscape schemes affect different brain regions differently.

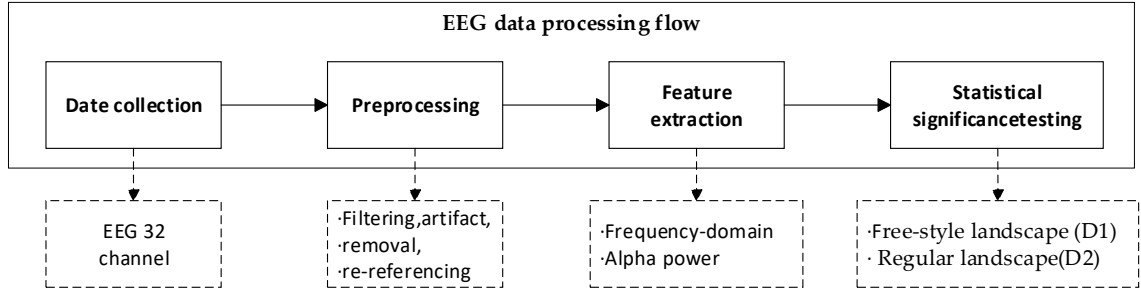

**Figure 4.** EEG data processing flow.

**Table 8.** Alpha values of some electrodes in some subjects and differences between the two groups.

| Electrode | | Subject1 | Subject2 | Subject3 | Subject4 | Subject5 | Subject6 | Subject7 |
|---|---|---|---|---|---|---|---|---|
| | | **M** | **M** | **M** | **M** | **M** | **M** | **M** |
| FP1 | D1 | 0.30 | −0.21 | 0.34 | 0.40 | −1.60 | −0.57 | −0.07 |
| | D2 | 0.23 | 0.18 | 0.58 | 0.03 | 0.58 | −0.32 | −0.29 |
| | D1–D2 | + | − | − | + | − | − | + |
| FPZ | D1 | 0.24 | −0.21 | 0.56 | 0.32 | −1.34 | −0.53 | −0.15 |
| | D2 | 0.20 | 0.28 | 0.78 | 0.08 | 0.53 | −0.26 | −0.26 |
| | D1–D2 | + | − | − | + | − | − | + |
| FP2 | D1 | 0.29 | −0.22 | 0.52 | −0.07 | −1.35 | −0.38 | −0.19 |
| | D2 | 0.15 | 0.29 | 0.97 | −0.35 | 0.51 | −0.27 | −0.28 |
| | D1–D2 | + | − | − | + | − | − | + |
| F7 | D1 | 0.20 | 0.10 | 0.22 | 0.07 | 0.97 | −0.98 | 0.01 |
| | D2 | 0.29 | −0.14 | 0.41 | −0.19 | −0.20 | −0.15 | −0.33 |
| | D1–D2 | − | + | − | + | + | − | + |
| F3 | D1 | 0.23 | −0.08 | 0.16 | −0.04 | 0.32 | −0.50 | −0.05 |
| | D2 | 0.12 | 0.09 | 0.30 | 0.13 | 0.19 | 0.16 | −0.03 |
| | D1–D2 | + | − | − | − | + | − | − |

Figure 5 shows the distribution of electrode locations for the 32 groups and their propensity for free-form or regular-form landscape. The results showed that more than half of the people who viewed D1 produced high alpha waves in the left frontal lobe (Fp1, Fp3), the left temporal lobe (Fc5, C3), and the central region electrodes (Fz, FC2, Cz), and that more people were more comfortable in this region for physiological perception of free-form landscape; when viewing the regular-form landscape, more people produced high alpha waves in the right frontal lobe (Fp2, F4, F8), the left parietal lobe (CP5, P3, CP1), and occipital electrodes (Pz, POz, O1), which means that the regular-form landscape is more likely to induce physiological comfort in these regions (see Figure 6). At the same time, we noticed that, similar to the F3 electrode in subject 7, there was no significant change in the α value before and after the video stimulation of the two sets of landscapes and, due to the different individual differences, we can only preliminarily determine the effect of the two sets of protocols on the physiological comfort in different brain regions from the perspective of the number of individuals.

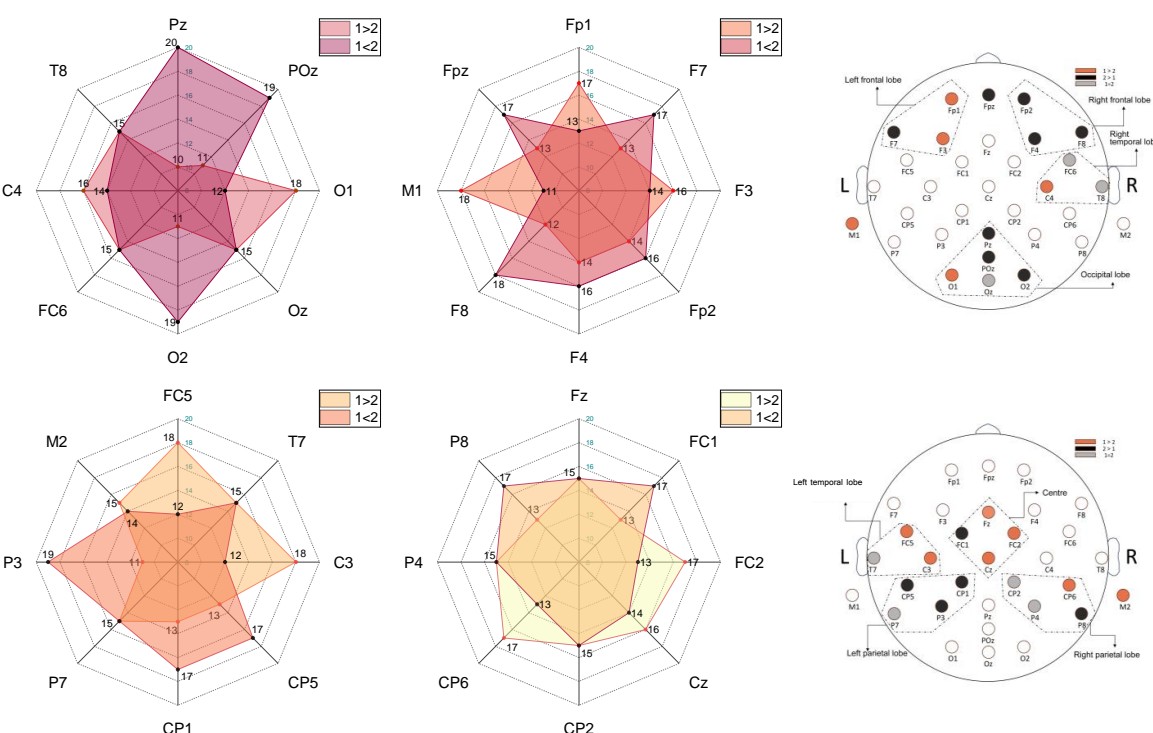

**Figure 5.** The number of people inclined toward the D1 and D2 plans for each electrode and the electrode area distribution.

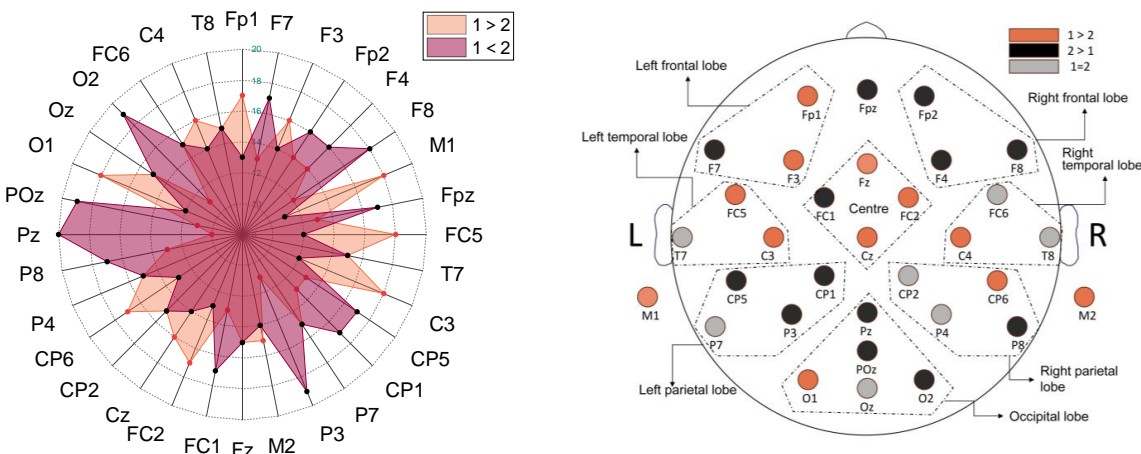

**Figure 6.** The number of people inclined toward D1 and D2 plans for all electrodes and the distribution of all electrode areas.

### 3.2.2. Comparison of α Values of Electrodes in Design D1 and D2

To examine the connection between the two landscapes and α values, we examined 32 sets of electrodes from subjects who were stimulated by the two protocols. We used the first 30 s of the experimental video as a baseline for the subjects' α values. To minimize errors caused by individual differences, we subtracted the baseline values from the actual experimental data. The α data from the 32 sets of electrodes did not follow a normal distribution, so we later tested them using a non-parametric test. Table 8 presents the average alpha values (excluding the baseline value) and the variability of the 32 groups of electrodes under the influence of the two protocols (see Figure 7) The results indicate that 18 out of the 32 groups of electrodes exhibited significant differences. Specifically, nine groups of electrodes (T8, POz, P4, Oz, O2, O1, M2, FC5, FC1) showed a significance level of $p < 0.000$, while six groups of electrodes (T7, Fz, F3, FC2, M1, C3) showed a significance

level of $p < 0.005$. Additionally, three groups of electrodes (Cz, P8, PZ) showed a significance level of $p < 0.05$.

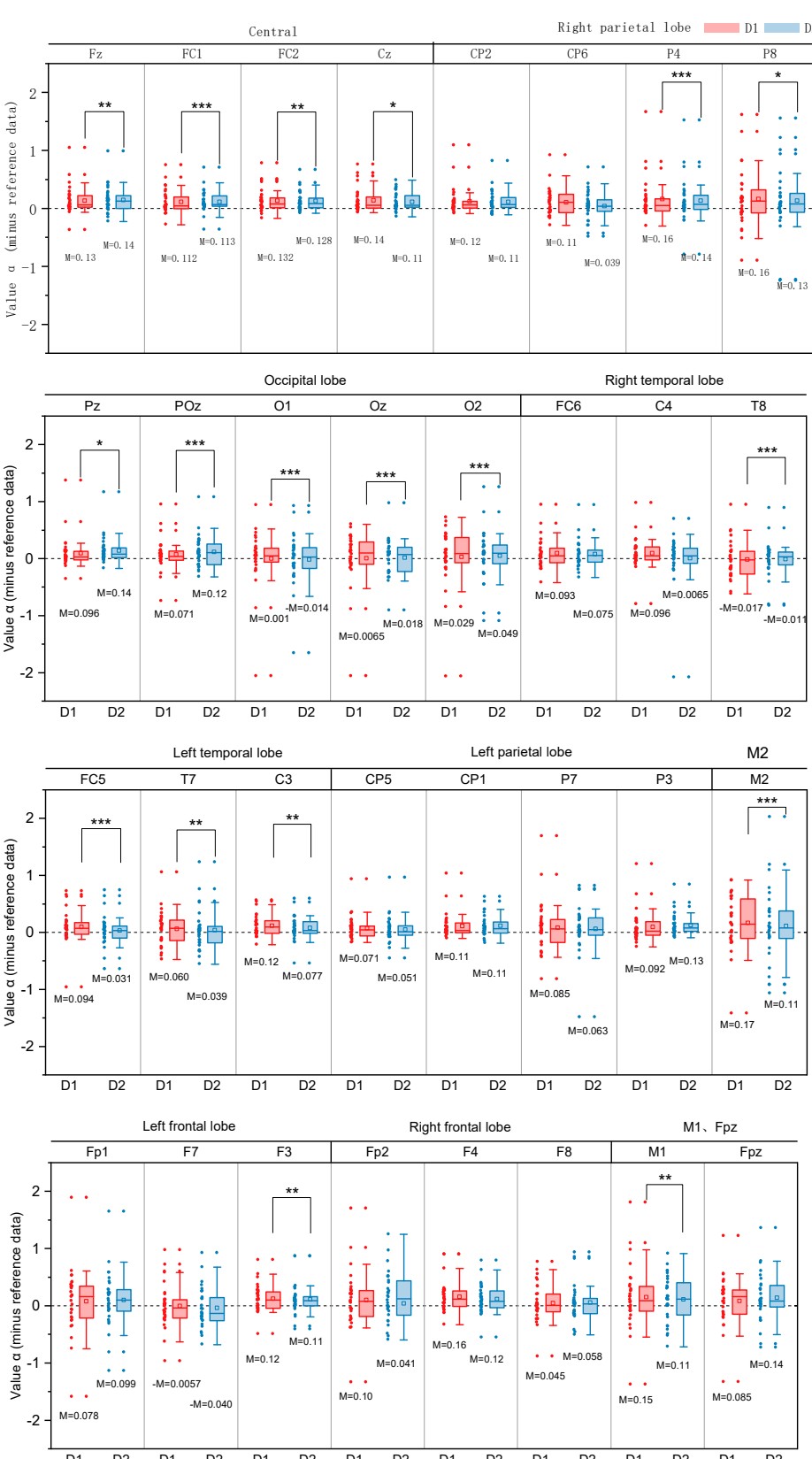

**Figure 7.** Average and difference analysis of electrode alpha waves in each brain region. Note: ***—$p \leq 0.001$, **—$p \leq 0.005$, *—$p \leq 0.05$.

Figure 7 displays the distribution of electrodes in the presence of discrepancies. It is evident that there were 18 sets of electrodes concentrated in specific areas: the occipital lobe (5 electrodes), the left temporal lobe (3 electrodes), the central region (4 electrodes), partially in the right parietal lobe (2 electrodes), the left frontal lobe (1 electrode), the right temporal lobe (1 electrode), and in the bilateral papillary positions of M1 and M2. From the perspective of brain region functions, the occipital lobe is involved in complex visual perception processes, including visual orientation, spatial frequency, and color discrimination. Additionally, the occipital lobe is also an important region for the perception of fatigue [19,20]. The left temporal lobe is responsible for utilizing visual memory to process sensory inputs and generate higher-level emotional awareness, and it influences emotional regulation [21–23]. The parietal lobe plays a crucial role in spatial perception and spatial orientation. It aids in the perception and understanding of the position, shape, and size of objects, as well as participating in spatial navigation and orientation abilities [24–26]. Thus, the electrode distributions with differences in alpha mean values indicate that the two landscapes had different effects on visual perception, color recognition, fatigue, brain mood, and spatial perception, and significantly different effects on visual perception and mood.

From the $\alpha$-mean values of each electrode in the two schemes, among the 18 groups of electrodes with significant differences in $\alpha$-mean values, 12 groups of electrodes had higher $\alpha$-values after viewing the free-form landscape D1, and 6 groups of electrodes had higher $\alpha$-mean values after viewing the regimented landscape D2. Thus, the landscape properties exhibited in the free-form landscape can cause more electrode $\alpha$ values to be higher, resulting in more physiological comfort.

Figure 8 shows the relationship between the size and distribution range of the mean $\alpha$ values of the two schemes for the 32 groups of electrodes. It can be seen that the three motors in the left temporal lobe and the two motors in the right parietal lobe had higher $\alpha$ values during D1 stimulation; the four electrodes in the occipital lobe area had higher $\alpha$ values during D2 stimulation; the central area FC2 and Cz electrodes had higher mean $\alpha$ values during the free-form landscape demonstration stimulation; and Fz and FC1 electrodes had higher mean $\alpha$ values during the regular-form landscape demonstration stimulation. The results show that the more natural plant configuration, zigzagging paths, hidden spaces, and other features in the free-form landscape can increase the $\alpha$-wave value of the left temporal lobe and the right parietal lobe, and, based on the functional characteristics of brain regions, the left temporal lobe plays an important role in the function of emotion regulation, which indicates that the free-form landscape has a more significant effect on the mood of the participants. For the regular-form landscape, regular landscape space, straight paths, geometric landscape vignettes, and other elements with strong visual stimulation can increase the $\alpha$-wave value of the occipital lobe area.

### 3.2.3. Individual Participant EEG Data Analysis

We discovered that free-form landscape and regular landscape had distinct effects on electrodes in different brain regions, which were concentrated in the left temporal lobe, right parietal lobe, and occipital lobe regions, based on the comparison of $\alpha$ values following stimulation of the two sets of schemes. Therefore, we used correlation analysis on the nearby electrodes in the region to confirm that the two sets of landscape schemes had regional impacts on the brain as opposed to the effects of individual electrodes. Figure 9 shows that when scheme D1 was stimulated, the correlation coefficient between two electrodes in the right parietal lobe and three electrodes in the left temporal lobe was more than 0.7, as was the correlation between adjacent electrodes in the occipital lobe region. In the meantime, the outcomes of scheme D2's stimulation closely matched those of scheme D1's stimulation, exhibiting a high degree of electrode correlation within the area. The findings imply that distinct brain regions are affected differently by the two landscape methods, and that these effects are more regionally biased than they are electrode-by-electrode.

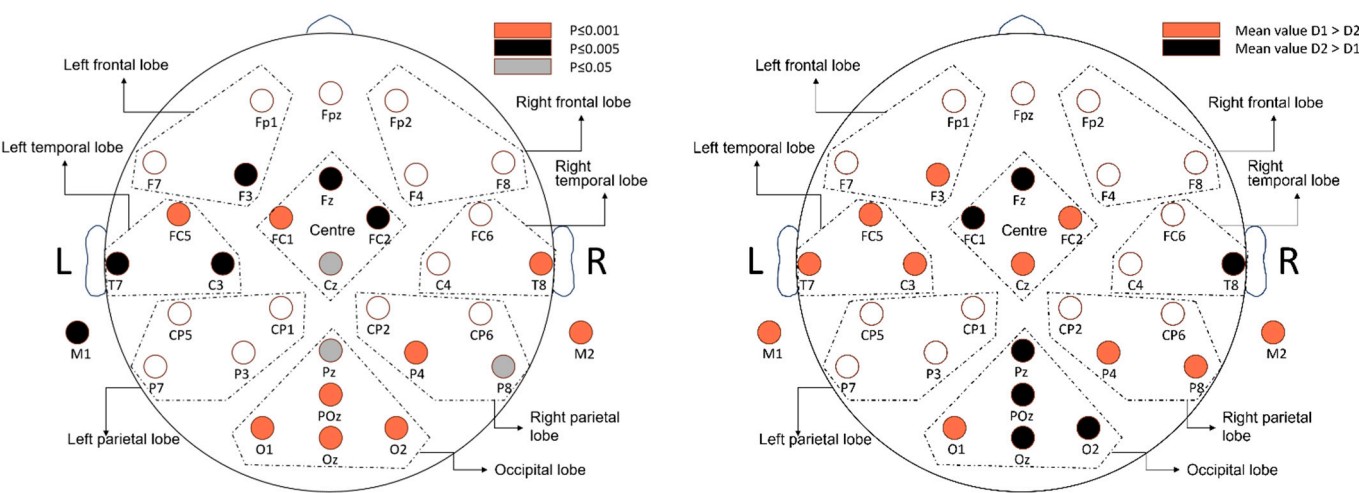

**Figure 8.** Comparison of the electrode difference value (*p* value) of each brain region and the α value of electrodes in each brain region of the two groups of plans.

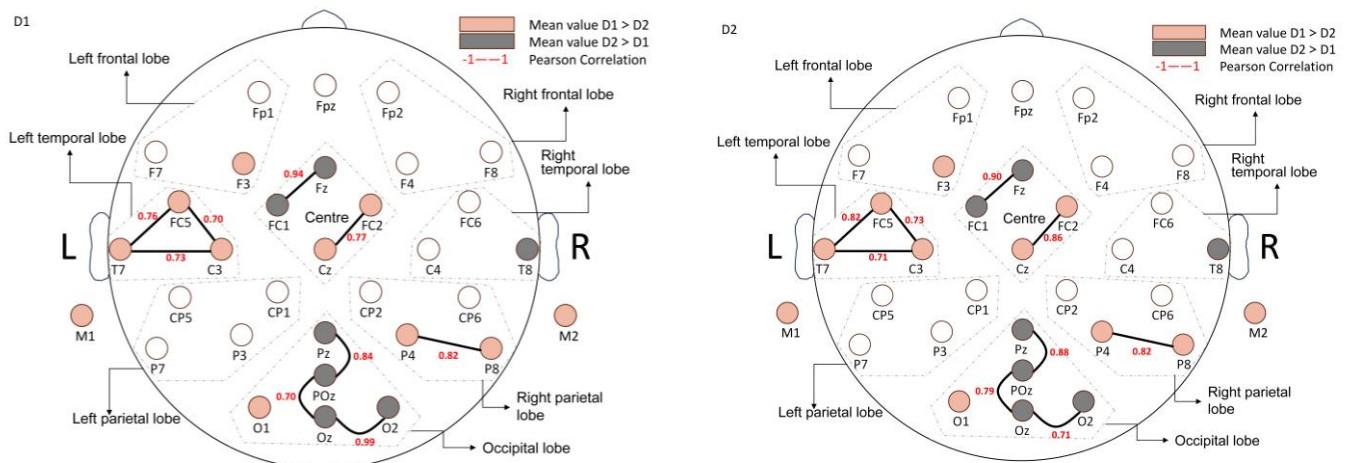

**Figure 9.** Analysis of alpha value correlation between electrodes of D1 and D2 designs.

### *3.3. Subjective Evaluation Results*

After the EEG test, subjective questionnaire scores were conducted on the landscapes of the two groups. The figures show the mean and standard deviation of scores for scenario D1 (see Figure 10) and the mean and standard deviation of scores for scenario D2 (see Figure 11). From an average point of view, the free-form landscape makes the subjects feel more natural, has a lower brightness perception, and is overall more hidden. The scale, rhythm, order, and other landscape characteristics of regular-form landscape are easier to perceive, and subjects think that regular landscapes are brighter.

The 15 factor indicators of the subjective questionnaires for the two groups were subjected to a differential analysis in SPSS. The data results indicate that the factors of perceived scale, rhythm perception, sense of security, and sense of dependency showed significant differences in the *t*-test, with *p*-values less than 0.05. However, the other 11 factors did not exhibit significant differences (see Table 9). The picture displays the average scores and differential analysis results for each factor between the two experimental conditions.

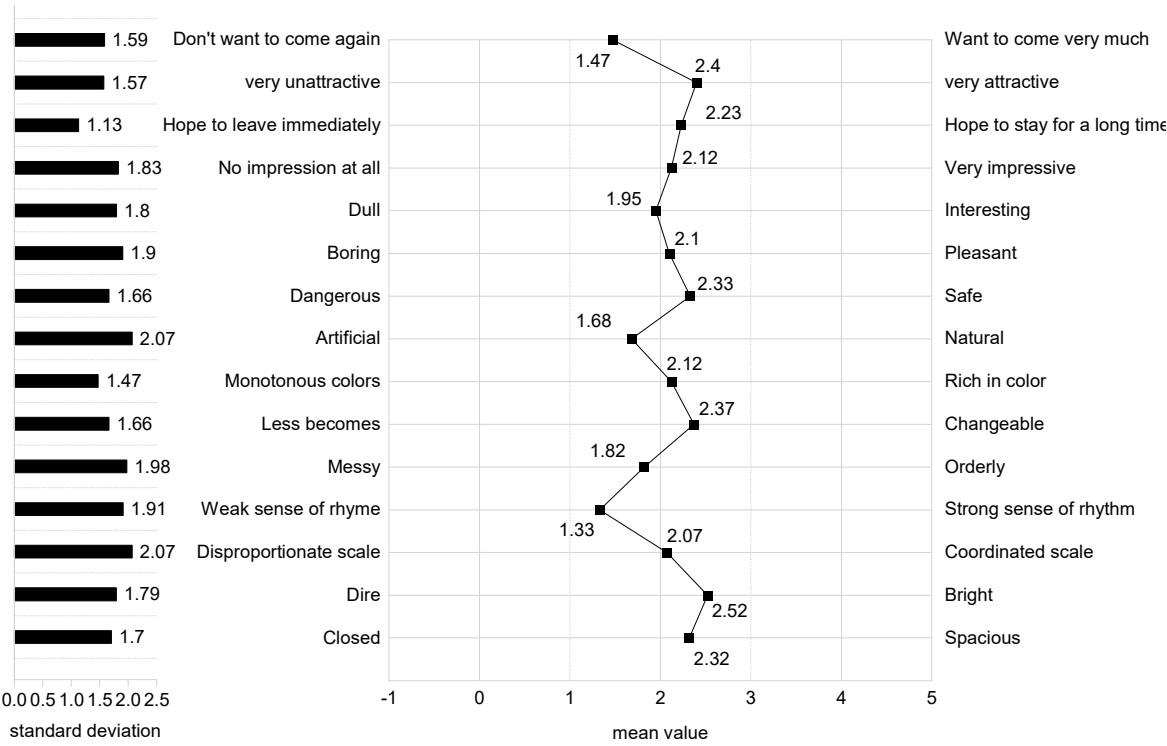

**Figure 10.** Plan D1 subjective evaluation score and standard deviation.

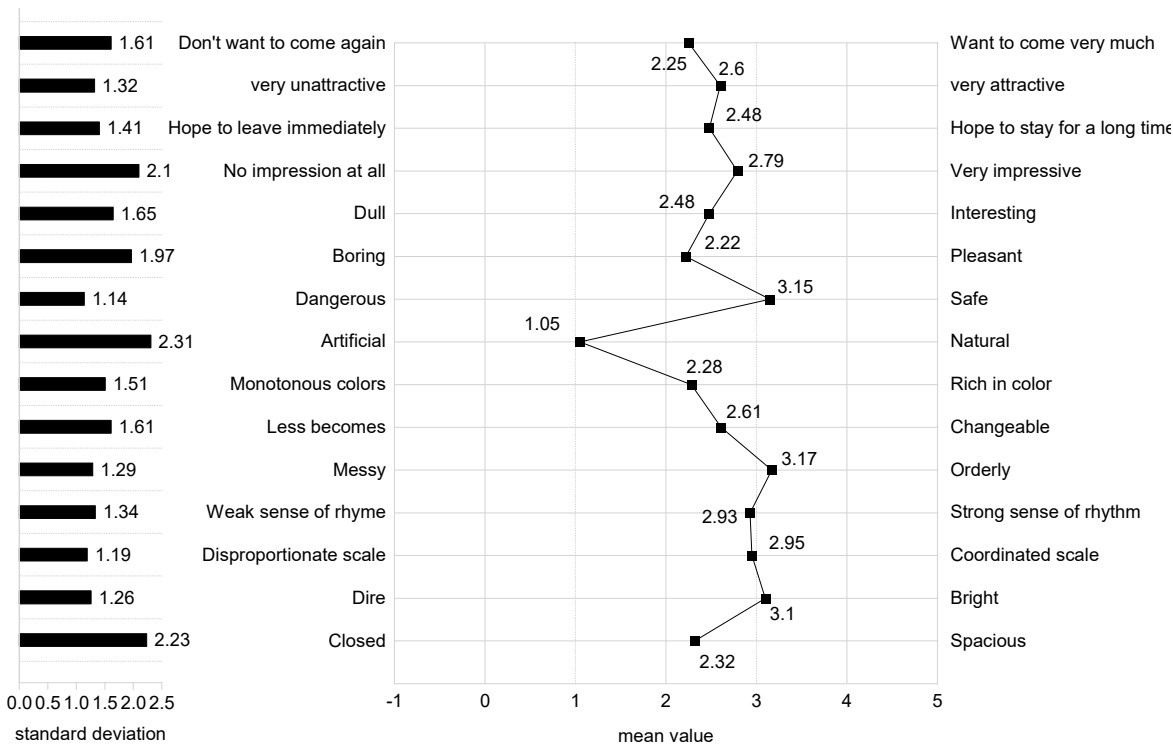

**Figure 11.** Plan D2 subjective evaluation score and standard deviation.

**Table 9.** Difference analysis of 15 groups of subjective evaluation factors (*p* value).

| Indicator | Space Perception | Brightness Perception | Scale Perception | Rhythm Perception | Order Perception | Variability | Colorfulness | Naturalness |
|---|---|---|---|---|---|---|---|---|
| U | 397,000 | 381,500 | 362,000 | 216,500 | 255,500 | 405,500 | 412,000 | 390,000 |
| W | 862,000 | 846,500 | 827,000 | 681,500 | 720,500 | 870,500 | 877,000 | 855,000 |
| Z | −0.801 | −1.051 | −1.339 | −3.516 | −2.963 | −0.666 | −0.576 | −0.896 |
| *p* | 0.423 | 0.293 | 0.181 | 0.000 | 0.003 | 0.505 | 0.564 | 0.370 |
| Indicator | Security | Pleasure | Interest | Dwell ability | Attractiveness | Impression | Dependent | / |
| U | 318,000 | 425,500 | 368,000 | 376,000 | 361,000 | 345,000 | 313,500 | / |
| W | 783,000 | 890,500 | 833,000 | 841,000 | 826,000 | 810,000 | 778,500 | / |
| Z | −2.007 | −0.368 | −1.230 | −1.118 | −1.367 | −1.592 | −2.046 | / |
| *p* | 0.045 | 0.713 | 0.219 | 0.263 | 0.172 | 0.111 | 0.041 | / |

The results showed that the subjects had significantly different perceptions of the landscape characteristics of the two groups of plans, and the order of magnitude of the difference based on subjective ratings is rhythmic sense > orderly sense. The subjective feelings of the two groups are obviously different, and the order is dependence > security (see Figure 12) The rhythm and order in landscape characteristics can be more easily felt in regular landscapes. Similarly, regular landscape can provide subjects with a higher sense of security and dependence.

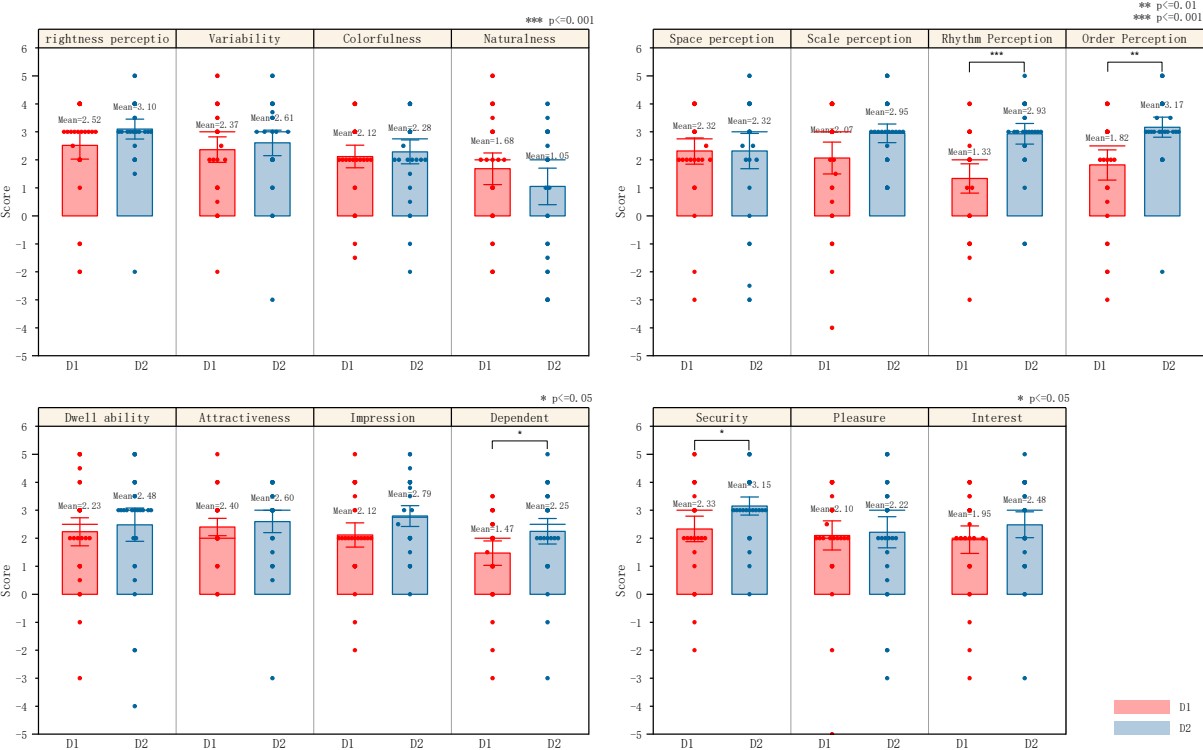

**Figure 12.** The mean scores and difference test results of 15 groups of subjective evaluation factors.

## 4. Correlation Analysis

*Correlation between Alpha Wave Values of Each Electrode and Factor Scores*

The article conducted a correlation analysis to determine the relationship between subjective factor scoring and EEG alpha values. Specifically, the difference values of alpha values from 18 groups of electrodes ($\alpha D1–\alpha D2$) and the difference values of subjective factor scoring (SD1–SD2) were chosen for this analysis (see Figure 13). The results indicated that there was a correlation between the differences in $\alpha$ mean values of O1 and Oz electrodes and the differences in rhythmic sense scoring. Similarly, the differences in $\alpha$ mean values of C3 electrodes were correlated with the differences in variability and color scoring. Additionally, the differences in $\alpha$ mean values of T7 and T8 electrodes were correlated with the differences in naturalness scoring.

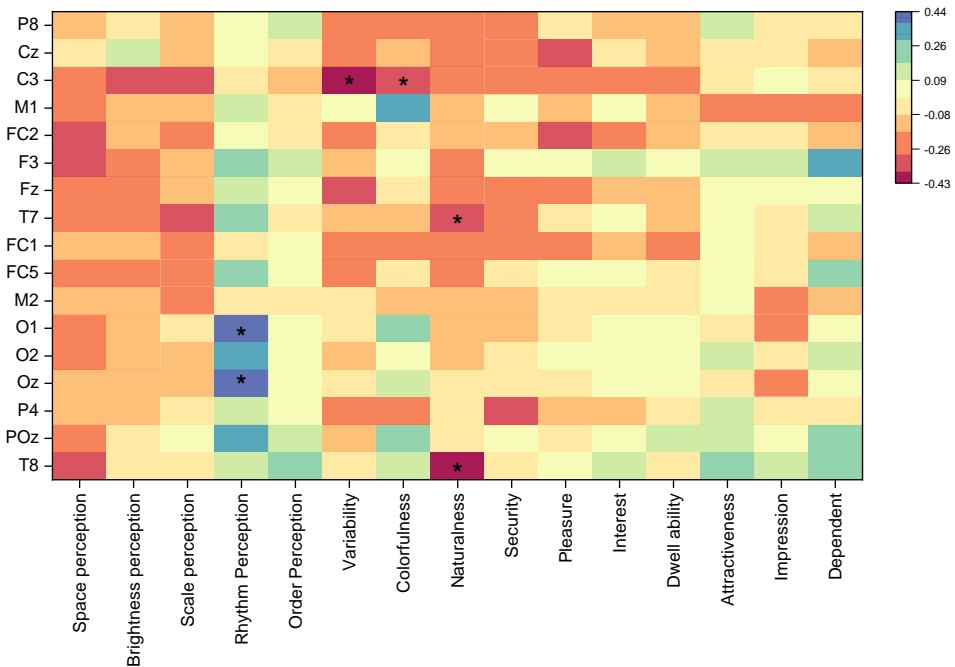

**Figure 13.** Subjective factor scores correlate with EEG $\alpha$. Note: *—$p \leq 0.05$.

From the analysis of the subjective factor scores for the two conditions, it is observed that free-form D1 received higher scores for naturalness while regular-form D2 scored higher in rhythm perception, variability, and chromaticity compared to condition D1. Analysis of the alpha values of the EEG electrodes saw that the subjects' O1, C3, and T7 electrodes had higher alpha averages with free-form landscape D1 stimulation and the T8 and O2 electrodes had higher alpha averages with regimented scheme D2 stimulation.

Figure 14 demonstrates the trend plot of the five EEG electrode alpha averages versus the factor score averages, and it can be seen that the O1 electrode alpha value in the occipital lobe area showed a positive correlation with the rhythmic sense scoring and the O2 electrode alpha value showed a negative correlation with the rhythmic sense scoring; the C3 electrode in the left temporal lobe showed a negative correlation with the degree of landscape change and the degree of colorfulness; the T8 electrode in the right temporal lobe showed a negative correlation with the sense of naturalness and the T7 electrode in the left temporal lobe showed a positive correlation with the sense of naturalness.

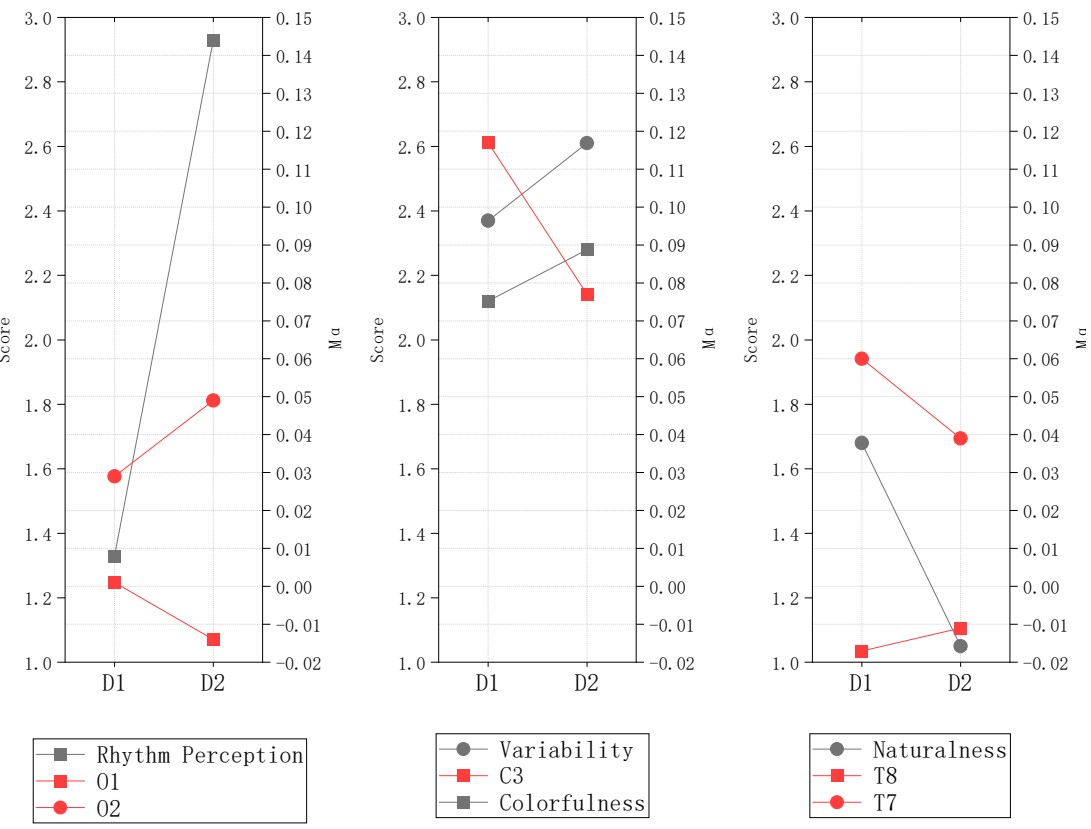

**Figure 14.** Comparison of α values and subjective factor scores of electrodes in design D1 and D2.

## 5. Conclusions

This paper uses EEG testing techniques and subjective evaluation methods to compare the comfort of people's experience of free-form and regimented landscapes, as well as to quantify and further explore the relationship between landscape characteristics and physiological comfort, leading to the following conclusions:

(1) From the EEG electrode α mean values, 12 groups of electrodes showed higher values during the D1 video stimulation of the free-form landscape, and 6 groups showed higher values during the D2 video stimulation of the regulated landscape. Overall, the free-form landscape is more capable of triggering human physiological comfort. Meanwhile, according to related studies, we know that free-form urban park scenes play an important role in people's psychological recovery [27], while viewing straight urban roads and square paved plazas will increase negative emotions [28,29]. Other studies have also shown that landscapes closer to natural forms have stronger stress recovery effects than regular artificial landscapes, and these studies provide an important basis for our results [30,31].

We observed significant variations in the impact of the two types of landscape designs on the comfort levels of different brain regions. Specifically, free-form landscape was found to induce greater physiological comfort in the left temporal lobe, right parietal lobe, and left frontal lobe. On the other hand, regular landscape was associated with higher physiological comfort in the occipital lobe regions and right temporal lobe at T8 electrodes. Both types of landscape designs were able to elicit physiological comfort in the central regions of the brain. The reason for this is due to the different functions of brain regions: the left temporal lobe, right parietal lobe, and left frontal lobe have emotional, sensory, and mental regulation roles, and the free-form landscapes, or more natural landscapes, can be relaxing and have a strong effect on people's emotional regulation [32–34], where the emotional regulation function of the free-form landscapes stimulates these brain regions effectively. Occipital and right temporal lobe regions are important parts of the brain for visual processing, while the significant role of regular landscape on visual stimuli induced

the vitality of these regions and showed higher $\alpha$-waves on EEG data [35–37], which is consistent with our findings.

(2) The results of the subjective questionnaire evaluation showed that there were significant subjective differences in participants' perceptions of the video stimuli between the two landscapes. From the perspective of subjective questionnaire scores, free-form landscapes were perceived as more natural and hidden, while the sense of order and rhythm of regular landscapes were better perceived by the participants.

(3) Using the EEG $\alpha$-wave as the basis for comfort evaluation, by analyzing the correlation between the EEG $\alpha$-value and subjective evaluation scoring, we obtained that the comfort of the right occipital lobe O2 region is positively proportional to the intensity of rhythm, and the comfort of the left occipital lobe O1 region is inversely proportional to the intensity of rhythm [38]; although the occipital lobe region is highly correlated with visual stimuli and landscapes with strong rhythms can trigger visual stimuli, the effect of this kind of stimuli on the left and right occipital alpha waves may be distinctly different. Comfort in the C3 region of the left temporal lobe was inversely proportional to variability and color, and comfort in the T7 region of the left temporal lobe was positively proportional to the sense of naturalness, which further suggests that freestyle landscapes are more capable of triggering physiological comfort in the C3 versus T7 regions, since freestyle landscapes were given lower scores for variability and color. We also obtained the result that comfort in the T8 region of the right temporal lobe was inversely proportional to the sense of nature [39,40]. The correlation results for the alpha mean sizes of the D1 and D2 electrodes were consistent with the performance of each electrode and the variations in subjective questionnaire scoring between the two regimens.

The results of this study show that there are obvious subjective cognitive differences between the subjects' perception of the rhythm and the degree of change in the free-form landscape and the regular landscape. Meanwhile, according to the analysis of the EEG test results, the EEG physiological data have a certain ability to perceive the characteristics of the free-form landscape and the regular landscape, and the free-form landscape can stimulate more areas of the brain and trigger higher alpha waves. The rhythmic size and degree of change in the landscape have a linear effect on the physiological comfort of certain brain regions, and the results of this study can provide a certain reference for the study of the relationship between landscape layout features and human physiological indicators.

At the same time, due to the number of subjects, individual differences, and other objective factors, this study also has certain limitations, but in today's increasingly prosperous EEG research, in the field of architecture and landscape design, the impact of human physiological indicators will be more fully explored.

**Author Contributions:** H.R.: conceptualization, methodology, supervision, review, editing; Z.Z.: conceptualization, methodology, software, validation, formal analysis, investigation, resources, data curation, writing, visualization, project administration; J.Z.: conceptualization, methodology, supervision, review, editing, funding acquisition; Q.W.: supervision, review. Y.W.: methodology, data curation. All authors have read and agreed to the published version of the manuscript.

**Funding:** This research was supported by the Basic Science Research Program through the National Research Foundation of Korea (NRF) funded by the Ministry of Education (RS-2023-00239818).

**Institutional Review Board Statement:** This study was conducted according to the guidelines of the Declaration of Helsinki and approved by the Institutional Review Board of Hebei University of Engineering (protocol code BER-YXY-2023031, approved 10 June 2023).

**Informed Consent Statement:** Informed consent was obtained from all subjects involved in this study. Written informed consent has been obtained from the subjects to publish this paper.

**Data Availability Statement:** The data presented in this study are available on request from the corresponding author. The data are not publicly available due to the large model files and large data volume.

**Conflicts of Interest:** The authors declare no conflicts of interest.

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
