# Peer review of "Electroencephalography (EEG)-Based Comfort Evaluation of Free-Form and Regular-Form Landscapes in Virtual Reality"

_applsci, doi:10.3390/app14020933_

Round 1
Reviewer 1 Report
Comments and Suggestions for Authors
I would like to congratulate the authors of this study, as what they present in this work is very interesting and complex because of the technology they have used.
The introduction is complete and clear, well thought out and helps to understand the topic of study. The objective is well written and precise.
With regard to the method, I have some doubts that I would like the authors to clarify in the paper. As can be seen in figure 1, the experimental subject is seated with the instruments in place (EEG and VR). Were the subjects free to stand up and move 360º freely around the space? I am aware that if so, there could be several problems: possibility of dizziness, if the subjects are not sufficiently adapted to the VR or interference and/or noise in the EEG due to the subject's movement. I think this is a drawback of the technology, but one that the authors have tried to solve in the best possible way and for which I congratulate them. But I think they should be included in the study.
With regard to the results section, I think it is very comprehensive and clearly set out. However, the discussion is very brief, leaving relevant aspects of the results found without discussing them with previous research. This is why I suggest discussing all the results found, even though I am aware of the small number of existing research in the authors' area of interest.
With these minor clarifications, the paper presents interesting results and is well executed.
Reviewer 2 Report
Comments and Suggestions for Authors
Major remarks
- Although the authors described the methodology and analysis in great detail, I am afraid a fundamental flaw is made with respect to the interpretation of EEG data. Throughout the paper, the authors compute alpha power for each single electrode and use count measures to find evidence in favor of more comfort for one of the two types of landscapes. However, this is not how you should analyze EEG data since the measured activity at eletrode locations is highly correlated between electrodes. Especially adjacent electrodes should correlate and if this is not the case, there is probably some more preprocessing to do.
So although the dataset might have some interesting possibilities, I can only reject this paper because this error is such a central part of the paper. I would advise to rework it.
Minor remarks
- line 16: "Relationship" doesn't need a capital letter.
- line 17. Try to avoid general statements like " the electrodes show significant differences". The brain is doing a billion things at the same time, so there will always be differences between conditions related to other variables than you are measuring. "Two-thirds of the electrodes' values are larger" is also non-informative if you don't specify the nature of these electrodes. I think it's better to talk about regions of interest in this context.
- line 25. Phrases like "feeling of the landscape have an impact on the comfort of electrodes" are really weird, electrodes don't have comfort. Try to rephrase sentences like these.
- line 57: "Generally, human brain waves have frequen-57 cies ranging from 1 to 30 Hz and can be divided into four bands: alpha (8-13 Hz), beta (14-58 30 Hz), theta, and delta waves [1]. " -> You can divide them in more than four bands
Comments on the Quality of English Languageshould be improved, especially the abstract
Reviewer 3 Report
Comments and Suggestions for Authors
The criteria for evaluating different landscapes were somewhat enigmatic. I understood that there were two films made. Were they produced by the same person? How should I interpret how they were made? Perhaps you investigated not the influence of the landscape but the influence of the production quality? It's problematic and very unclear. The criteria for the "random" and "regular" plans are not clear. What indicators did you use for the evaluation? Is it your subjective opinion or some measurable parameter?
Please comment on the research procedure regarding the head movement of the study participant. Was head movement recorded, and was the impact of movement on the quality of EEG signals examined? I would appreciate more details about the setup of the cap with the HMD. I am concerned that the photograph shows a rather uncomfortable way of conducting the experiment. The position of the participant raises significant concerns, especially if someone is to participate in the experiment for over 6 minutes.
I request information on how subjective data were practically collected (I lack information about the scale used, only the content of the question is provided). I emphasize that the experiment should be described in a way that allows it to be replicated by other researchers. This is a fundamental requirement for objective scientific work, and I urge you to provide this information. Please specify the criteria by which you believe the sample size for the adopted surveys is sufficient.
Round 2
Reviewer 2 Report
Comments and Suggestions for Authors
I don't think you answered my concern regarding the inter-correlatedness of electrode activity and the way you interpret the activations at the level of the electrode.
Reviewer 3 Report
Comments and Suggestions for Authors
Overall, I am pleased with the improvements made.
